# Bovine Lactoferrin and Hen Ovotransferrin Affect Virulence Factors of Acute Hepatopancreatic Necrosis Disease (AHPND)-Inducing *Vibrio parahaemolyticus* Strains

**DOI:** 10.3390/microorganisms11122912

**Published:** 2023-12-02

**Authors:** Marieke Vandeputte, Margaux Verhaeghe, Lukas Willocx, Peter Bossier, Daisy Vanrompay

**Affiliations:** 1Laboratory of Immunology and Animal Biotechnology, Department of Animal Production and Aquatic Ecology, Faculty of Bioscience Engineering, Ghent University, 9000 Ghent, Belgium; marieke.vandeputte@ugent.be (M.V.); margauve.verhaeghe@ugent.be (M.V.); lukas.willocx@ugent.be (L.W.); 2Laboratory of Aquaculture & Artemia Reference Center, Department of Animal Production and Aquatic Ecology, Faculty of Bioscience Engineering, Ghent University, 9000 Ghent, Belgium; peter.bossier@ugent.be

**Keywords:** *Vibrio parahaemolyticus*, AHPND, acute hepatopancreatic necrosis disease, bovine lactoferrin, ovotransferrin, transferrin

## Abstract

Acute Hepatopancreatic Necrosis Disease (AHPND), a highly destructive shrimp disease, has inflicted severe setbacks on the shrimp farming industry worldwide. As the use of antibiotics is discouraged due to emerging antibiotic-resistant bacteria and the pollution of ecosystems, there is a pressing demand for novel, sustainable alternatives. Hence, the influence of bovine lactoferrin (bLF) and hen ovotransferrin (OT), two natural antimicrobial proteins, on the growth of three AHPND-causing *Vibrio parahaemolyticus* (*Vp*) strains (M0904, TW01 and PV1) was examined. Additionally, we explored their potential to affect selected *Vp* virulence factors such as biofilm formation, swimming and swarming, cell surface hydrophobicity, and activity of released lipases and caseinases. Lag phases of all bacterial growth curves were significantly prolonged in the presence of bLF or OT (1, 5 and 10 mg/mL), and bLF (5 and 10 mg/mL) completely inhibited growth of all strains. In addition, bLF or OT significantly reduced biofilm formation (all tested bLF and OT concentrations for *Vp* M0904 and *Vp* PV1), bacterial swimming motility (0.5 mg/mL bLF and OT for *Vp* M0904 and *Vp* TW01; 1 mg/mL bLF and OT for all strains), cell surface hydrophobicity (for all strains, all bLF and OT concentrations tested except for 0.125 mg/mL OT for *Vp* PV1) and lipase activity (1 mg/mL bLF and OT for all strains and 0.5 mg/mL bLF and OT for *Vp* PV1). These promising in vitro results suggest that bLF and/or OT might be used as novel agents for combating AHPND and warrant further research to elucidate the underlying mechanisms of action to fully unlock their potential for AHPND disease management.

## 1. Introduction

In the last decades, the global increase in demand for seafood has led to the intensification of aquaculture production systems. This intensification has introduced a range of challenges, including the emergence of infectious disease outbreaks. One such disease that has a profound impact on the shrimp farming industry worldwide is Acute Hepatopancreatic Necrosis Disease (AHPND). This disease was identified in China in 2009 for the first time under the name Early Mortality Syndrome (EMS), and it has since spread to other Asian countries such as Thailand, Vietnam and Malaysia, but also Mexico and the United States [1,2,3]. Still, AHPND continues to inflict large mortalities and losses in shrimp farms, where it develops after approximately eight days post stocking, with severe mortalities (up to 100%) occurring within 20 to 30 days [2]. The number one causative agent is the Gram-negative *Vibrio* (*V.*) *parahaemolyticus*. However, only strains harboring the pVA1 plasmid (69 kb) which carries the *pirAB* toxin genes are able to induce AHPND [4,5]. More recently, other *Vibrio* species have been identified with a plasmid homologous to pVA1 harboring the *pirAB* genes, such as *V. harveyi* [6], *V. campbellii* [7,8,9], *V. owensii* [10] and *V. punensis* [11]. Central to the pathogenicity of AHPND in shrimp is the PirAB binary toxin, yet other factors that contribute to virulence are biofilm formation, flagellar motility, extracellular proteases and lipases, and type III and VI secretion systems [2,12]. Antibiotics have been used and, at times, misused for decades to address bacterial diseases in aquaculture [13], resulting in the pollution of aquatic ecosystems and the emergence of antibiotic-resistant bacteria, including *V. parahaemolyticus* and other *Vibrio* species [14,15,16]. Consequently, many countries have implemented prohibitions and restrictions on the use of antibiotics, creating a pressing demand for novel, sustainable alternatives [13,17]. In the present study, we investigated the potential of two such alternatives, bovine lactoferrin (bLF) and hen ovotransferrin (OT), belonging to the family of transferrins (TFs), a class of multifunctional glycoproteins naturally found in both vertebrates and invertebrates. This family has conserved characteristics and all primarily serve to control iron levels in biological fluids [18]. Beyond their iron transport function, TFs have gained significant attention for their involvement in immune defense mechanisms such as antibacterial, -fungal, -viral, -parasitic, antioxidant and anti-inflammatory activities [17,19,20]. Both TFs used in this study, bLF and OT, are easily accessible and produced on an industrial scale. Structurally, they both consist of two homologous lobes (N and C), each able to reversibly bind one Fe^3+^ cation along with one CO_3_^2−^ [18,21]. In native conditions, they generally contain 15 to 20% iron and are called holo-transferrins, while when they contain less than 5% iron, they are called apo-TF. Both TFs are widely used in various industries, such as food preservation and nutritional supplements, and in numerous medications due to their non-toxic and ecological features [17,22]. Important antibacterial modes of action of the TFs are iron sequestration, rendering iron inaccessible to invading pathogens and interaction with the bacterial cell membrane through binding to lipopolysaccharide (LPS), porins or other outer membrane proteins (OMPs) in Gram-negative bacteria, resulting in perturbed membrane stability and structure [17,18,21]. Furthermore, they are proteolytic enzymes, able to degrade important virulence factors of, for example, enterohemorrhagic and enterotoxigenic *Escherichia* (*E.*) *coli* (EHEC and ETEC) [23,24]; can disrupt the bacterial Type III secretion system (T3SS) [25]; and have been shown to inhibit the attachment and cell entry of obligatory intracellular *Chlamydia* (*C.*) [26].

Historically, the antibacterial effects of bLF and OT were first studied using various bacteria of human and mammals including *E. coli* strains [23,24], *Chlamydia* spp. [26,27] and *Bacillus* spp. [28,29]. Regarding *Vibrio* spp., inhibition of growth by bLF was observed for human pathogenic *V. parahaemolyticus* strains (non-AHPND) [30] and *V. cholerae* [31], and bLF was able to work synergistically with antibiotics to inhibit growth in *V. fluvialis*, *V. alginolyticus*, *V. vulnificus* and *V. furnissii* [31]. Furthermore, treatment of *V. cholerae* with bLF resulted in severe membrane damage, such as the occurrence of bacterial protrusion and filamentation [31]. Recently, multiple advantages of using TFs have been proposed for fish [17] and crustaceans [32,33]. Studies with TFs or TF derivatives in shrimp are scarce but rendered positive results such as enhanced growth, reduced mortalities, decreased disease outbreaks and stimulated immune responses [32,33]. In this research, we explored the potential of bLF and OT for use in aquaculture against *Vibrio parahaemolyticus* induced AHPND. In vitro experiments on three different AHPND+ *V. parahaemolyticus* strains were performed to investigate the effect of bLF and OT on the growth of the bacteria and on selected virulence factors. To our knowledge, this is the first study investigating the in vitro effects of both bLF and OT on AHPND-causing *Vibrio parahaemolyticus* strains. The findings of this study provide valuable insights into the antimicrobial effects of the transferrins on this significant shrimp pathogen. This holds promise for the potential application of these compounds as bioactive feed additives in shrimp aquaculture. Such applications could significantly influence various aspects of the aquaculture industry, including food security, economic stability, employment and social welfare.

## 2. Materials and Methods

### 2.1. Transferrins

Stock solutions of ovotransferrin (OT) (Bioseutica, Zeewolde, The Netherlands) and bovine lactoferrin (bLF) (Ingredia, Arras, France) were prepared in filtered (0.2 µm), autoclaved artificial seawater (FAASW) containing 35 g/L of Instant Ocean synthetic sea salt (Aquarium systems, Sarrebourg, France), and filtered and sterilized (0.22 µm). According to the manufacturers, OT was 4.8% iron saturated, while bLF was 9% iron saturated.

### 2.2. Bacterial Strains and Growth Conditions

Three different *V. parahaemolyticus* strains were used in this study: *Vp* M0904 was isolated in Northwestern Mexico (received from A.C. Mazatlàn unit of Aquaculture), *Vp* TW01 in Southern Thailand and *Vp* PV1 in China (both received from Robins McIntosh). All three strains were isolated from AHPND-affected shrimp and were previously confirmed to have the pirAB genes by PCR with PirAB2020 primers (Phiwsaiya et al., 2017). Depending on the experiment, the bacteria were cultured in either Marine Broth (MB) (Carl Roth, Karlsruhe, Germany), Tryptic Soy Broth (TSB) (Carl Roth, Karlsruhe, Germany) with addition of 15 g/L extra NaCl, or Luria–Bertani (LB) 35 medium (10 g/L tryptone (Carl Roth, Karlsruhe, Germany), 5 g/L yeast extract (Carl Roth, Karlsruhe, Germany) and 35 g/L NaCl (Carl Roth, Karlsruhe, Germany) in distilled water). Cultures were grown overnight at 28 °C with shaking at 140 rpm before further use.

### 2.3. PirAB Production by Vp Test Strains

PirAB production was verified by Western blotting. Briefly, overnight cultures were centrifuged for 15 min at 4000× *g* (4 °C), and supernatants were filtered (0.22 µm) to produce cell-free supernatant (CFS). Toxins in CFS were concentrated using ultra-centrifugal filters with a molecular weight cut-off of 10 kDa. Samples were treated with 0.01 M β-mercaptoethanol (5 min, 95 °C) and run on a 4–15% Mini-Protean TGX Stain-Free Protein Gel (Bio-Rad, Hercules, CA, USA) (15 min, 130 V; 45 min, 150 V). Western blotting was performed (60 min, 100 V) and PVFD membranes (Bio-Rad, Hercules, CA, USA) were incubated with in-house-produced polyclonal rabbit anti-PirA or polyclonal rabbit anti-PirB antibodies (1:7500) followed by a polyclonal goat anti-rabbit IgG (HRP) antibody (1:10,000; Genscript, Piscataway, NJ, USA). Clarity Western ECL Substrate (Bio-Rad, Hercules, CA, USA) was used for detection.

### 2.4. Effect of bLF and OT on Bacterial Growth

To test the bacteriostatic/bactericidal activity of OT and bLF on *V. parahaemolyticus*, the effect on the growth curves of the three different strains was assayed in flat-bottomed 96-well plates (VWR International, Radnor, PA, USA). The strains were first grown overnight in TSB (+1.5% NaCl) and centrifuged, and the pellets were resuspended in fresh broth to obtain four different bacterial concentrations (10^1^, 10^3^, 10^5^ and 10^7^ CFU/mL). All bacterial concentrations were supplemented with 10, 5, 1, 0.1, 0.01, 0.001 and 0 mg/mL of bLF or OT and were tested in triplicate in 96-well plates. Pure broth was used as blanks. Growth of the cultures was monitored for 24 h at 28 °C by hourly absorbance measurements at 550 nm with a spectrophotometer (Infinite^®^ 200 PRO, Tecan, Mechelen, Belgium). Data were analyzed using a modified Gompertz model [34], represented by the following formula:Y = Y0 + (YM − Y0) * exp(−exp(K * (lag − X)/(YM − Y0) + 1))

Parameters Y0 and YM are the starting and maximum population, respectively, expressed as a value of optical density (OD) at 550 nm; lag is expressed in h and is indicative of the duration of the lag phase; and K is a parameter expressed as reciprocal time units (h^−1^). A visual representation can be found in Figure 1.

### 2.5. Effect of bLF and OT on Bacterial Virulence Factors

The effect of bLF and OT (0, 0.125, 0.250, 0.5 and 1 mg/mL) on bacterial biofilm formation, swimming and swarming motility, cell surface hydrophobicity (CSH) and activity of bacterial lipases and caseinases was investigated. Overnight cultures were grown in LB35 for all experiments except for the biofilm formation assay (MB).

#### 2.5.1. Biofilm Formation

The effect of the two TFs on biofilm formation was assayed in 96-well plates using a crystal violet staining. Overnight cultures were diluted with fresh broth to OD_550_ 0.1 and supplemented with 0, 0.125, 0.250, 0.5 or 1 mg/mL bLF or OT. Four replicates of 200 µL were introduced to the wells of a 96-well plate and incubated for 48 h at 28 °C without shaking. The wells were washed three times with 200 µL PBS to remove the bacterial cell cultures, dried for 30 min and the remaining bacteria were fixated with 200 µL methanol (99%, Thermo Fisher Scientific, Cambridge, UK). After 2 h, the methanol was removed and the plates were left to dry overnight. Fixed cells were stained with 200 µL of 0.1% crystal violet (Carl Roth, Karlsruhe, Germany) for 20 min. Plates were carefully washed with tap water to remove the excess of crystal violet and dried again for 2 h. Finally, 95% ethanol (200 µL/well) (Thermo Fisher Scientific, Cambridge, UK) was added to dissolve the crystal violet and the absorbance was measured at 570 nm with a spectrophotometer.

#### 2.5.2. Swimming and Swarming Motility

For the swimming and swarming motility tests, soft agar plates of LB35 were prepared with 0.2% and 0.6% agar (Biokar Diagnostics, Pantin, France), respectively, comprising the TF concentrations mentioned before. Overnight cultures were diluted to OD_550_ 0.1, and 2 µL was used for the inoculation of five replicate plates per condition. Culture plates were incubated in the upright (swimming) or inverted (swarming) positions for maximum 20 h at 28 °C, after which the diameters of the colonies were measured.

#### 2.5.3. Cell Surface Hydrophobicity (CSH)

Furthermore, the effect of the TFs on CSH was measured with the bacterial adherence to hydrocarbons (BATH) test based on (12). Briefly, 25 µL of overnight culture was brought first in fresh LB35 broth and incubated with the TF test concentrations for 24 h at 28 °C and 140 rpm. Thereafter, cultures were centrifuged at 11,000× *g* for 15 min, and pellets were washed with PBS and once again centrifuged at 11,000× *g* for 7 min. The resulting cultures were diluted in PBS to an OD_550_ value of 0.5 (=A_0_). A total 4 ml of these suspensions were added to 1 mL of p-xylene (Honeywell, Charlotte, NC, USA), mixed and then left to separate for 30 min at room temperature. Finally, triplicates of the watery phase (200 µL) were added to the wells of the 96-well plates and OD_550nm_ was measured (=A_i_). PBS was used as a blank. Hydrophobicity (%) was determined as follows:Hydrophobicity (%) = ((A_0_ − A_i_)/A_0_) * 100
(<20% = not hydrophobic, 20–50% = moderate, >50% = strong)

#### 2.5.4. Extracellular Enzyme Activity

Lipase activity was measured in quintuplicate on LB35 agar plates (15 g/L agar) comprising 1% Tween80 (Merck, Darmstadt, Germany) and the previously mentioned TF test concentrations. Overnight cultures were diluted to OD_550_ 0.1 and 2 µL was used for the inoculation. Plates were incubated inverted for three days at 28 °C before measurements. The caseinase activity of the supernatants from the three strains was tested as described by [12]. Briefly, overnight cultures were 1:100 diluted in LB35 and incubated for 24 h at 28 °C and 140 rpm with the TF test concentrations. The cultures were then centrifuged for 15 min at 15,000× *g* and 75 µL of the supernatant was mixed with 125 µL azocasein (2% *w*/*v*; Megazyme, Wicklow, Ireland) solution and incubated for 30 min at 37 °C. To end proteolysis, 660 µL trichloroacetic acid (10%) (Merck, Darmstadt, Germany) was added. The solutions were kept at −20 °C for 30 min to precipitate the remaining azocasein and subsequently centrifuged for 10 min at 10,000× *g*. A total 600 µL of the supernatants was added to 700 µL of NaOH (1 M) (VWR International, Radnor, PA, USA), three replicates of 200 µL were placed in 96-well plates and the OD440 nm was measured. LB35 was used as blank.

#### 2.5.5. Statistical Analysis

Data were analyzed with GraphPad Prism. For the growth curves, statistical analysis of the derived parameters YM and lag was performed by one-way ANOVA with Dunnett multiple comparison testing, where the mean of each condition was compared with the mean of the control. Data of all other tests were also analyzed with one-way ANOVA with Dunnett multiple comparison testing to compare the mean of each condition with the mean of the respective control. For both swimming and swarming motility, the data were first log-transformed.

## 3. Results

### 3.1. PirAB Production by Vp Test Strains

The production of PirA and PirB was successfully demonstrated for all test strains (Figure 2).

### 3.2. Effect of bLF and OT on Bacterial Growth

The growth curves for all test strains in the presence or absence of transferrins are shown in Figure 3 (for 10^1^ CFU/mL) and Figure 4 (for 10^3^ CFU/mL). A summary of the parameters YM and lag, derived from non-linear fitting to the Gompertz model, can be found in Table 1.

For *Vp* M0904, amounts of 10 and 5 mg/mL bLF were able to completely prevent bacterial growth when starting with 10^1^ CFU/mL. Furthermore, compared with the control, there was a significantly longer lag phase; thus, growth inhibition was observed when using 0.01 and 1 mg/mL bLF and 1, 5 and 10 mg/mL OT. When starting with 10^3^ CFU/mL of *Vp* M0904, 1, 5 and 10 mg/mL bLF and 5 and 10 mg/mL OT were able to significantly postpone growth. Interestingly, 0.001, 0.01 and 1 mg/mL bLF were able to significantly lower maximal growth (YM) when starting with 10^1^ CFU/mL, while 0.1, 1, 5 and 10 mg/mL OT significantly increased maximal growth. Similar effects were observed when using 10^3^ CFU/mL. For *Vp* TW01, 10 and 5 mg/mL bLF completely prevented bacterial growth, regardless of whether the cultures were started with 10^1^ or 10^3^ CFU/mL. Moreover, 0.001, 0.1 and 1 mg/mL bLF and 1, 5 and 10 mg/mL OT significantly postponed bacterial growth as shown by the longer lag phase at a starting concentration of 10^1^ CFU/mL. For the 10^3^ CFU/mL cultures, bacterial growth was significantly delayed when using 1 mg/mL bLF and 5 and 10 mg/mL OT. For a starting concentration of 10^3^ CFU/mL in the presence of 1 mg/mL bLF, the maximal bacterial growth rate (YM) was significantly lower compared with the controls. On the other hand, regardless of the starting concentration, maximal bacterial growth values (YM) were significantly higher than those for the controls when using 0.1, 1, 5 and 10 mg/mL OT. For *Vp* PV1, regardless of the starting concentration, 5 and 10 mg/mL bLF again completely inhibited growth while 1 mg/mL bLF significantly prolonged the lag phase for both bacterial starting concentrations. The same was observed for 1, 5 and 10 mg/mL OT. Regardless of the starting concentration, the maximal bacterial growth rate (YM) was significantly lower when adding 1 mg/mL bLF or 1, 5 and 10 mg/mL OT. However, a significant increase in YM was observed when using 0.01 mg/mL bLF.

The growth curves and parameters for the cultures that were started with 10^5^ and 10^7^ CFU/mL can be found in Appendix A). In general, similar observations can be made as those for the experiments starting with 10^1^ and 10^3^ CFU/mL. Lag phases were significantly prolonged, with the highest concentrations of transferrins compared with the control when starting with 10^5^ CFU/mL and with differences becoming less pronounced with 10^7^ CFU/mL. Regardless of the starting populations, addition of the highest concentration of OT resulted again in a significantly higher YM for *Vp* M0904 and TW01, and in a significantly lower YM for *Vp* PV1. Addition of bLF had a varying effect on YM, resulting in a higher YM for 10 mg/mL in *Vp* M0904 (10^5^ and 10^7^ CFU/mL) and 0.1, 5 and 10 mg/mL in *Vp* PV1 (10^7^ CFU/mL); lowering YM at concentrations higher than 1 mg/mL in *Vp* PV1 (10^5^ CFU/mL); and having no effect on TW01.

### 3.3. Effect of bLF and OT on Bacterial Virulence Factors

#### 3.3.1. Biofilm Formation

In the absence of transferrins, all test strains were able to create biofilms (Figure 5). However, biofilm production by *Vp* M0904 was more pronounced (OD_570_ > 0.4) than that for *Vp* TW01 and *Vp* PV1 (both (OD_570_ = 0.1). All transferrin concentrations significantly inhibited biofilm formation by *Vp* M0904 or *Vp* PV1. Transferrins had no effect on biofilm formation by *Vp* TW01.

#### 3.3.2. Swimming and Swarming Motility

All test strains showed a comparable swimming motility in the absence of transferrins. The swimming motility of *Vp* M0904 and *Vp* TW01 was significantly reduced by both bLF and OT at 0.5 and 1.0 mg/mL (Figure 6 and Figure 7). For *Vp* PV1, this was only the case at 1 mg/mL of both transferrins. Strangely, for *Vp* PV1, lower concentrations of OT (0.125 and 0.25 mg/mL) significantly increased swimming. Swarming by our test strains was not observed without transferrins or when only low concentrations (<0.5 mg/mL) of transferrins were present in the medium. However, at the highest (1 mg/mL) bLF and OT concentration used, swarming by all test strains significantly increased (Figure 6 and Figure 7). Depending on the strain, 0.5 mg/mL of one or both transferrins also significantly increased swarming.

#### 3.3.3. Cell Surface Hydrophobicity (CSH)

Cell surfaces of all test strains were strongly hydrophobic in the absence of transferrins (CSH > 50%). Bovine LF always significantly lowered CSH for all test strains. All OT concentrations used for *Vp* M0904 significantly lowered CSH (Figure 8). For *Vp* PV1 and *Vp* TW01, CSH was significantly reduced when using ≥0.25 and ≥0.5 mg/mL OT, respectively.

#### 3.3.4. Extracellular Enzyme Activity

All test strains displayed caseinase activity in the absence of transferrins, and depending on the strain and the transferrin, addition of ≤0.125 or ≤0.250 mg/mL significantly augmented caseinase activity (Figure 9). All test strains displayed lipase activity, albeit minimal, in the absence of transferrins or when low concentrations of transferrins were added to the agar. However, once higher concentrations (1 mg/mL for *Vp* M0904 and *Vp* TW01 and 0.5 and 1 mg/mL for *Vp* PV1) were included in the agar, no precipitation and, thus, no lipase activity was observed anymore (Figure 10).

## 4. Discussion

This is the first report assessing the antibacterial activity of bovine lactoferrin (bLF) and ovotransferrin (OT) on AHPND-causing *V. parahaemolyticus* strains. Previous research has explored their antimicrobial properties against various pathogens, revealing bacteriostatic, bactericidal and toxin-degrading activities [17,23,30,31]. Specifically, bLF has previously demonstrated inhibitory effects on the growth of various *Vibrio* species, including human pathogenic (non-AHPND) *V. parahaemolyticus* [30], *V. cholerae*, *V. fluvialis*, *V. vulnificus* and *V. alginolyticus* [31]. Additionally, preliminary tests suggest that bLF can positively impact fish and shrimp health, with benefits such as an enhanced growth, reduced mortalities, decreased disease outbreaks and stimulated immune responses [17,32,33], making them a potential antibiotic alternative for use in aquaculture. As a crucial first step, this study investigated the in vitro effects of bLF and OT on AHPND-causing *V. parahaemolyticus* strains from different geographical regions. 

The growth curve assay revealed that both bLF and OT exhibited antibacterial effects on AHPND-positive *V. parahaemolyticus* strains. The highest concentrations of bLF completely inhibited growth for 24 h in all three strains, indicating a bactericidal effect. Additionally, a bacteriostatic effect was observed for the highest concentrations of OT and some concentrations of bLF since growth was significantly delayed. Intriguingly, a strain-specific effect on the maximal population concentration (YM) was observed for OT, as YM was increased in strains *Vp* M0904 and TW01 for some OT concentrations, but not in *Vp* PV1. A difference in effectivity between bLF and OT was also observed in various E. coli strains by Dierick, Van der Weken, Rybarczyk, Vanrompay, Devriendt and Cox [23]. They found that bLF concentrations of 5 mg/mL and higher could inhibit growth, whereas OT had no impact on growth. Furthermore, Leon-Sicairos, Canizalez-Roman, de la Garza, Reyes-Lopez, Zazueta-Beltran, Nazmi, Gomez-Gil and Bolscher [30] observed a concentration-dependent activity of bLF on *V. parahaemolyticus* strains, with the lowest concentration (10 µM) inhibiting growth and higher concentrations (30 and 40 µM) having no effect on growth anymore. Altogether, the antibacterial effect of transferrins seem to be depending on the transferrin type, bacterial species and strain, and used concentrations. The observed antibacterial effects likely result from the combined effect of iron sequestration by the TFs, and direct damage to the bacterial cell membrane, as previously described [17,18,21]. However, the observation that two out of three strains eventually exhibited increased growth with OT indicates potential adaptation mechanisms. In each context, bacteria must adequately sense, process and invoke functional responses to environmental changes to ensure their survival and proliferation, and they have evolved strategies to do so [35]. Given the essential role of iron for bacterial growth, bacteria have evolved strategies to thrive in low-iron environments. *V. parahaemolyticus*, for instance, employs siderophores (Vibrioferrin) to scavenge iron from host iron-binding proteins such as TF, LF and ferritin [36]. While prior research has shown vibrioferrin’s ability to capture iron from human TF, but not from human LF [37], its interaction with bLF or OT has not been explored to the best of our knowledge. In addition to siderophores, bacterial outer membranes can directly bind transferrins through receptors [38], although *V. parahaemolyticus* lacks identified transferrin receptors. It is plausible that the strains tested in our study produce siderophores or TF receptors specific to OT but not bLF, accounting for our observed outcomes. Several genes involved in vibrioferrin production have been identified in the genomes of the three *Vp* strains (unpublished results). Furthermore, it is important to note that *V. parahaemolyticus* produces numerous extracellular proteases, which could potentially degrade TFs into smaller peptides or nucleic acids, providing a nutrient source for the pathogen [39]. In this case, the high OT concentrations could provide an extra nutrient source to the bacteria, leading to the higher maximal growth observed for strains *Vp* M0904 and TW01 in the presence of high OT concentrations.

To further understand the antibacterial mechanisms of bLF and OT, their impact on a set of virulence factors was studied. All concentrations of both bLF and OT inhibited biofilm formation in strains *Vp* M0904 and PV1, while *Vp* TW01 remained unaffected. For the control conditions, there was considerable variation in biofilm-formation ability among the three strains, with *Vp* M0904 consistently producing more biofilm. This variability among other strains of *V. parahaemolyticus* has been observed in previous studies too [40,41,42]. Notably, a reduction in biofilm formation has been observed in other bacteria following LF treatment, including Pseudomonas aeruginosa, Klebsiella pneumoniae and E. coli, and was mainly attributed to membrane damage [21]. Biofilms play a critical role in shielding microorganisms from environmental fluctuations, antibiotics and disinfectants, making them essential for *Vibrio* species’ adaptation and survival in aquatic ecosystems like aquaculture [41,42,43]. Consequently, a decrease in biofilm formation may indicate a reduction in bacterial virulence. Furthermore, both tested transferrins affected flagella-mediated swimming and swarming motility, with swimming being inhibited and swarming induced at high transferrin concentrations. These motility processes are pivotal in the pathogenesis of *V. parahaemolyticus*, facilitating initial attachment and infection by overcoming repulsive forces between bacterial cells and host surfaces [43]. Consequently, inhibiting swimming motility may reduce the bacteria’s infectiousness. Moreover, lower swimming motility has been linked to reduced biofilm formation capacity in *V. parahaemolyticus*, a correlation validated by our results [12,41,44]. Conversely, the induction of swarming motility is suggested to be a result of the iron depletion caused by the transferrins, as *Vibrio* species tend to activate swarming in nutrient-deficient environments [41]. Additionally, low-iron conditions have been shown to stimulate both swarming and T3SS regulons in other *V. parahaemolyticus* strains [45].

Another virulence factor assessed in our study is CSH. Our results demonstrate that all CSH values exceeded 50%, indicating a robust hydrophobicity of the bacterial cell surfaces, consistent with previous research [40,41]. However, the introduction of TFs led to a significant reduction in CSH. Considering the documented binding capabilities of bLF and OT to LPS, porins and other outer membrane proteins, it is plausible that this interaction may be concealing hydrophobic regions of, e.g., LPS [46]. Consequently, cells with a lower CSH will have a higher tendency to remain in aqueous environments rather than attaching to hydrophobic (a)biotic surfaces [46,47]. Therefore, CSH plays a critical role in the initial cell attachment step of biofilm formation [12,43], and the observed decrease in CSH may contribute to the diminished biofilm formation observed in our strains exposed to transferrins.

Lastly, we examined the effect of bLF and OT on extracellular enzyme activity—more specifically, lipases and caseinases. Without transferrins, all three strains showed caseinase and lipase activity, as expected for *V. parahaemolyticus* strains [40]. However, the presence of the transferrins significantly stimulated caseinase activity, while lipase activity diminished at high transferrin concentrations. These extracellular enzymes play a crucial role in host tissue damage, facilitating nutrient acquisition and tissue penetration by the pathogens [12,39]. Consequently, the reduction in lipase activity could potentially diminish bacterial virulence. Conversely, the induction of caseinase activity might have the opposite effect. Since caseinase production in certain *Vibrio* species is suggested to be regulated by quorum sensing (QS) [48], further investigation into the impact of transferrins on QS regulators in *V. parahaemolyticus* is warranted.

## 5. Conclusions

In summary, our findings collectively underscore the substantial impact of both bLF and OT on AHPND-causing *V. parahaemolyticus* strains. The introduction of the transferrins resulted in reduced CSH and decreased swimming motility, which can be directly correlated with the observed inhibition of biofilm formation. This inhibition is of particular significance, given the challenges posed by *Vibrio* species biofilm formation in the context of aquaculture treatment. Moreover, the growth inhibition observed holds promise for future applications. Nevertheless, it is noteworthy that two out of three strains exhibited the ability to recover and even attain higher maximal growth in the presence of OT, suggesting that this compound may be less suitable for combating AHPND-causing *V. parahaemolyticus*.

In conclusion, our study provides valuable insights into the antibacterial effects of bLF and OT on AHPND-causing *V. parahaemolyticus* strains, giving new perspectives for their use in shrimp aquaculture. The observed antibacterial effects, coupled with transferrins’ reported immune-stimulating properties in other studies, make them promising candidates for further in vivo research related to their use as antimicrobials in shrimp aquaculture against AHPND.

## Figures and Tables

**Figure 1 microorganisms-11-02912-f001:**
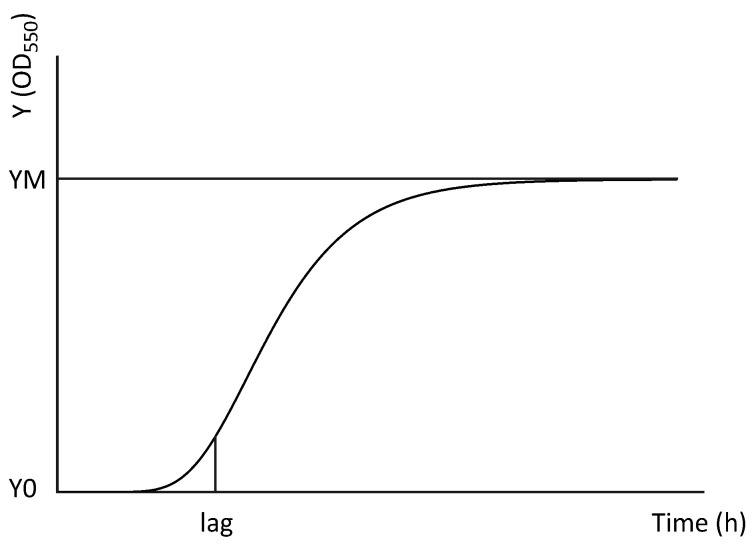
Visual representation of the modified Gompertz growth function (Y = Y0 + (YM − Y0) * exp(−exp(K * (lag − X)/(YM − Y0) + 1)). Y0 and YM represent the starting population and maximum population (upper asymptote), respectively, and are both expressed as values of optical density (OD_550_). Lag (expressed in h) represents the time point when the exponential bacterial growth actually begins.

**Figure 2 microorganisms-11-02912-f002:**
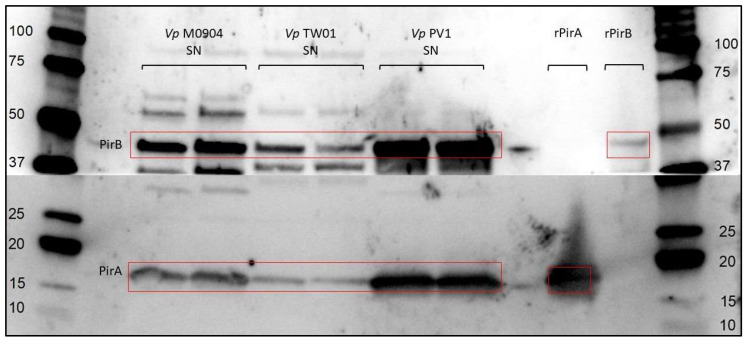
Western blot of concentrated supernatant of *Vp* M0904, PV1 and TW01. Blot was cut after protein transfer and incubated separately with either anti-PirA or anti-PirB antibodies. Recombinantly produced PirA and PirB (rPirA/B) were used as controls.

**Figure 3 microorganisms-11-02912-f003:**
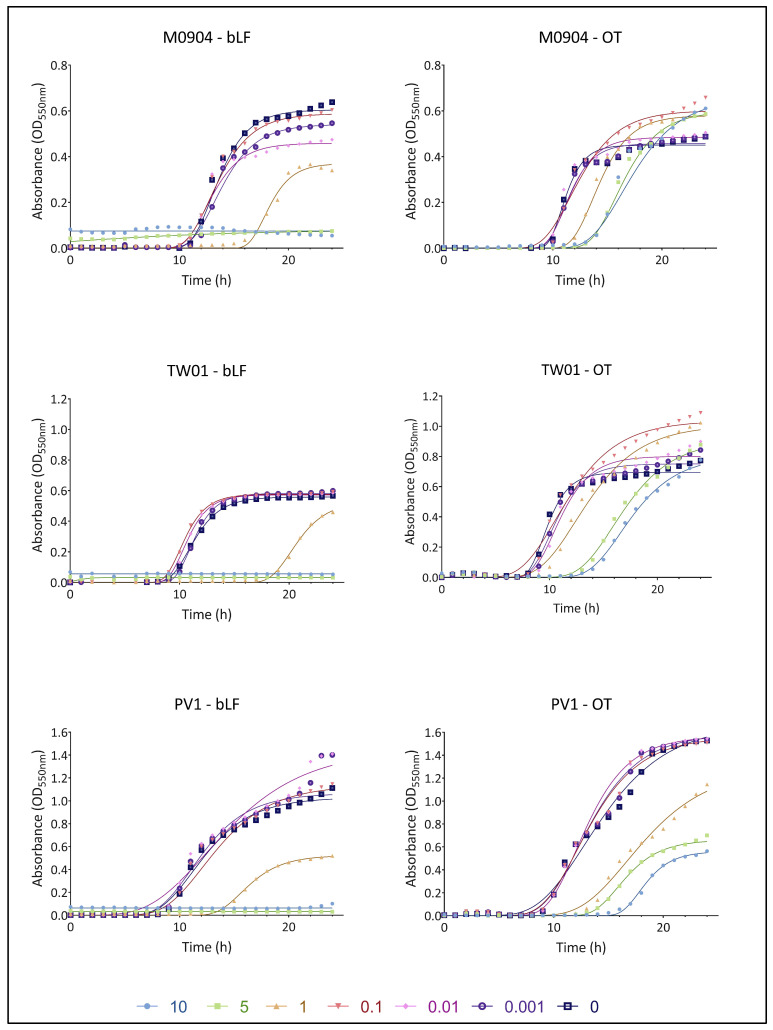
Growth curves of *Vp* M0904, *Vp* TW01 and *Vp* PV1, with a starting concentration of 10^1^ CFU/mL, and incubation with bLF and OT at concentrations 0, 0.001, 0.01 0.1, 1, 5 and 10 mg/mL. Individual points represent the mean OD_550_ (n = 3); curves are the non-linear fitting to the modified Gompertz model.

**Figure 4 microorganisms-11-02912-f004:**
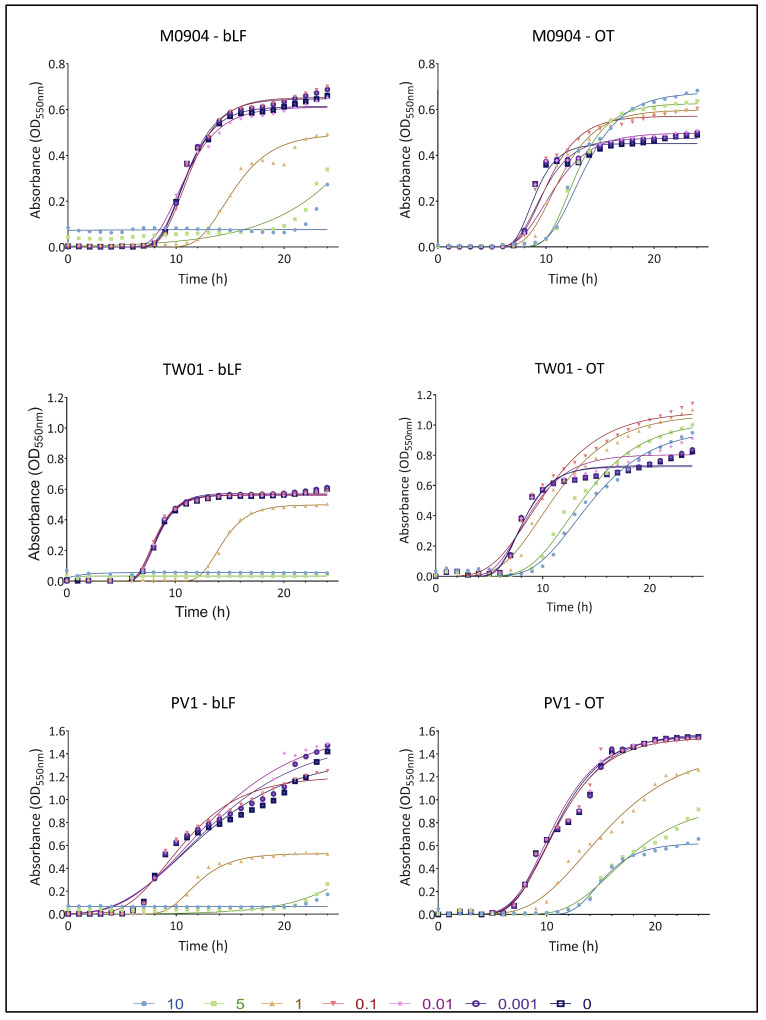
Growth curves of *Vp* M0904, *Vp* TW01 and *Vp* PV1, with a starting concentration of 10^3^ CFU/mL, and incubation with bLF and OT at concentrations 0, 0.001, 0.01 0.1, 1, 5 and 10 mg/mL. Individual points represent the mean OD_550_ (n = 3); curves are the non-linear fitting to the modified Gompertz model.

**Figure 5 microorganisms-11-02912-f005:**
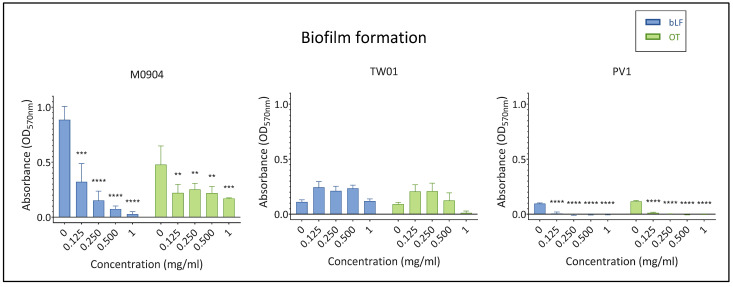
Effect of bLF and OT on biofilm formation. Mean absorbance at OD_570nm_ and 95% confidence interval are shown on the graphs. Significant differences between transferrin treatment and relative control (0 mg/mL) are indicated by asterisks: ** *p* < 0.01, *** *p* < 0.001, **** *p* < 0.0001.

**Figure 6 microorganisms-11-02912-f006:**
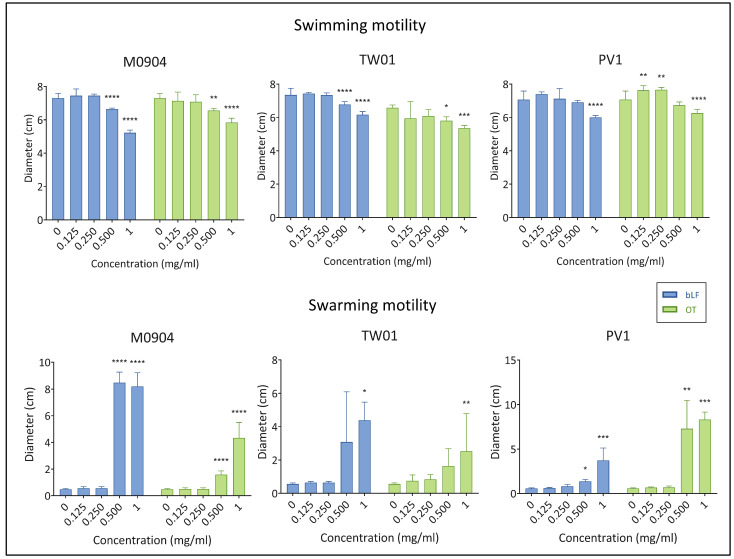
Effect of bLF and OT on swimming and swarming motility. Mean diameters of the swimming or swarming colonies and 95% confidence interval are shown on the graphs. Significant differences between transferrin treatment and relative control (0 mg/mL) are indicated by asterisks: * *p* < 0.05, ** *p* < 0.01, *** *p* < 0.001, **** *p* < 0.0001.

**Figure 7 microorganisms-11-02912-f007:**
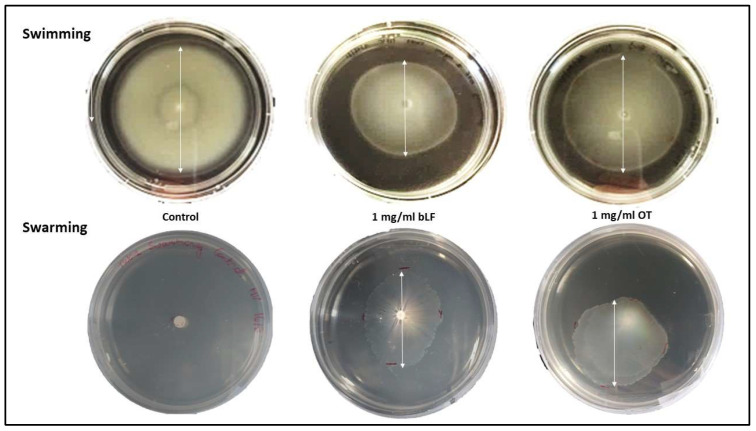
Pictures of the motility plate assays for *Vp* TW01 at 0 and 1 mg/mL of bLF or OT. Diameters of the swimming or swarming colony are illustrated by a white arrow. In the presence of transferrins, swimming and swarming decreased or increased, respectively, compared with the controls.

**Figure 8 microorganisms-11-02912-f008:**
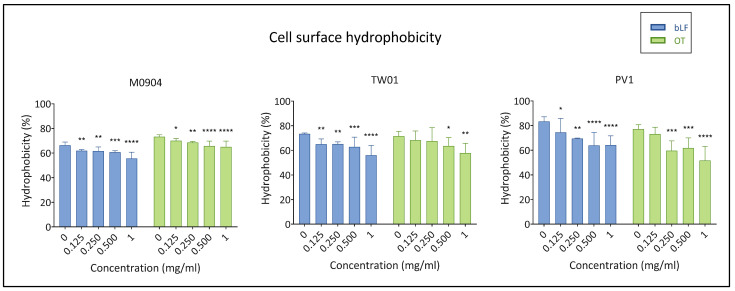
Effect of bLF and OT on cell surface hydrophobicity (CSH). Mean CSH (%) and 95% confidence interval are shown on the graphs. Significant differences between transferrin treatment and relative control (0 mg/mL) are indicated by asterisks: * *p* < 0.05, ** *p* < 0.01, *** *p* < 0.001, **** *p* < 0.0001.

**Figure 9 microorganisms-11-02912-f009:**
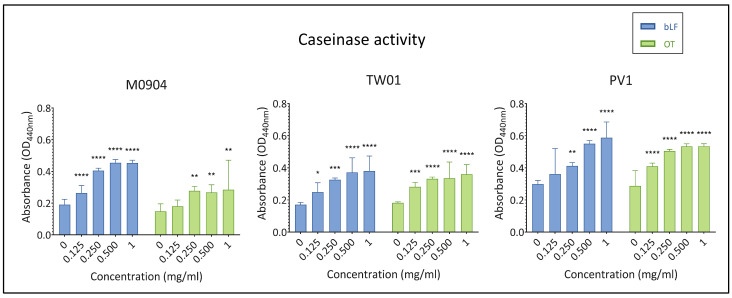
Effect of bLF and OT on caseinase activity of the supernatant. Mean absorbance at OD440 nm and 95% confidence interval are shown on the graphs. Significant differences between transferrin treatment and relative control (0 mg/mL) are indicated by asterisks: * *p* < 0.05, ** *p* < 0.01, *** *p* < 0.001, **** *p* < 0.0001.

**Figure 10 microorganisms-11-02912-f010:**
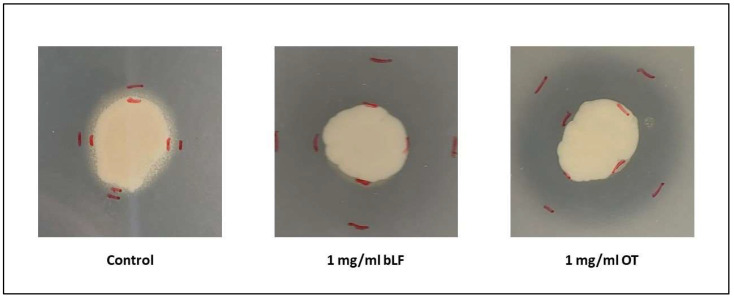
Pictures of the lipase activity assay for *Vp* TW01 at 0 and 1 mg/mL of bLF or OT. A precipitation zone around the colony indicates extracellular lipase activity by the bacteria.

**Table 1 microorganisms-11-02912-t001:** A summary of the bacterial growth parameters of *Vp* M0904, *Vp* TW01 and *Vp* PV1 with starting populations of 10^1^ and 10^3^ CFU/mL, derived from the modified Gompertz model. YM represents the maximum population, expressed as values of optical density (OD_550_); Lag represents the length of the lag time in h. Between brackets, significant differences between transferrin treatment and relative control (0 mg/mL) are indicated by asterisks: * *p* < 0.05, ** *p* < 0.01, *** *p* < 0.001, **** *p* < 0.0001. Non-significant differences are indicated by ‘ns’. (/: modified Gompertz model was not able to calculate growth parameters due to too long lag phase or completely flat curve.)

*Vp* M0904					
	Treatment (mg/mL)	bLF		OT	
		YM	Lag	YM	Lag
**10^1^ CFU/mL**	**10**	0.075 (****)	/	0.656 (****)	13.375 (****)
	**5**	0.080 (****)	/	0.598 (****)	13.500 (****)
	**1**	0.370 (****)	16.382 (****)	0.580 (****)	11.757 (****)
	**0.1**	0.588 (ns)	10.700 (ns)	0.601 (****)	9.125 (ns)
	**0.01**	0.458 (****)	10.546 (*)	0.485 (ns)	9.509 (ns)
	**0.001**	0.541 (****)	11.084 (ns)	0.457 (ns)	9.527 (ns)
	**0**	0.605	11.142	0.450	9.625
**10^3^ CFU/mL**	**10**	/	/	0.675 (****)	10.176 (****)
	**5**	/	/	0.626 (****)	10.068 (****)
	**1**	0.493 (****)	11.972 (****)	0.598 (****)	8.160 (**)
	**0.1**	0.653 (*)	8.644 (ns)	0.571 (****)	7.481 (ns)
	**0.01**	0.610 (ns)	7.907 (ns)	0.498 (***)	7.478 (ns)
	**0.001**	0.647 (*)	8.396 (ns)	0.474 (ns)	7.213 (ns)
	**0**	0.613	8.475	0.451	7.098
***Vp*** **TW01**					
	**Treatment (mg/mL)**	**bLF**		**OT**	
		**YM**	**Lag**	**YM**	**Lag**
**10^1^ CFU/mL**	**10**	/	/	0.801 (*)	13.694 (****)
	**5**	/	/	0.908 (***)	12.802 (****)
	**1**	0.530 (ns)	18.256 (****)	1.008 (****)	9.078 (ns)
	**0.1**	0.576 (ns)	8.664 (*)	1.036 (****)	7.392 (ns)
	**0.01**	0.572 (ns)	8.977 (ns)	0.801 (ns)	8,450 (ns)
	**0.001**	0.581 (ns)	9.588 (*)	0.751 (ns)	8.248 (ns)
	**0**	0.558	9.148	0.695	8.051
**10^3^ CFU/mL**	**10**	/	/	0.977 (****)	9.181 (****)
	**5**	/	/	1.027 (****)	8.723 (***)
	**1**	0.498 (****)	12.184 (****)	1.070 (****)	6.106 (ns)
	**0.1**	0.564 (ns)	6.439 (ns)	1.088 (****)	5.123 (ns)
	**0.01**	0.560 (ns)	6.532 (ns)	0.804 (ns)	5.394 (ns)
	**0.001**	0.574 (ns)	6.643 (ns)	0.731 (ns)	5.711 (ns)
	**0**	0.561	6.624	0.724	5.825
***Vp*** **PV1**					
	**Treatment (mg/mL)**	**bLF**	**OT**		
		**YM**	**Lag**	**YM**	**Lag**
**10^1^ CFU/mL**	**10**	/	/	0.561 (****)	16.184 (****)
	**5**	/	/	0.661 (****)	13.651 (****)
	**1**	0.523 (****)	13.489 (****)	1.290 (**)	12.527 (****)
	**0.1**	1.137 (ns)	8.645 (ns)	1.559 (ns)	8.977 (ns)
	**0.01**	1.447 (***)	7.303 (ns)	1.564 (ns)	9.411 (ns)
	**0.001**	1.065 (ns)	8.107 (ns)	1.586 (ns)	9.058 (ns)
	**0**	1.029	8.191	1.668	8.613
**10^3^ CFU/mL**	**10**	/	/	0.621 (****)	12.428 (****)
	**5**	/	/	0.964 (****)	11.916 (****)
	**1**	0.528 (***)	9.032 (***)	1.429 (ns)	8.674 (***)
	**0.1**	1.204 (ns)	5.386 (ns)	1.544 (ns)	6.848 (ns)
	**0.01**	1.615 (ns)	5.110 (ns)	1.554 (ns)	6.760 (ns)
	**0.001**	1.534 (ns)	4.845 (ns)	1.585 (ns)	6.687 (ns)
	**0**	1.375	4.584	1.565	6.939

## Data Availability

The data presented in this study are available in this article and in Appendix A.

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
