# Peer review of "Bovine Lactoferrin and Hen Ovotransferrin Affect Virulence Factors of Acute Hepatopancreatic Necrosis Disease (AHPND)-Inducing *Vibrio parahaemolyticus* Strains"

_microorganisms, 2023, doi:10.3390/microorganisms11122912_

Round 1
Reviewer 1 Report
Comments and Suggestions for Authors
In my opinion, the manuscript has scientific relevance for shrimp aquaculture. In addition, it is well written and presented. The use of lactoferrin and ovotransferrin on shrimp suggests novel and sustainable antimicrobials for aquacualture diseases.
Reviewer 2 Report
Comments and Suggestions for Authors
Over the past few years, the interest in Vibrio parahaemolyticus has been based on strategies for antimicrobial effects. In my opinion, this is a well-planned study and has a series of experiments that build on the fundamental aspects and the phenotype. Overall, I think this work is publishable with a few minor comments.
In the introduction section: The author should add more discussion about the importance and significance of this study.
In line no 142: Could you explain Figure 1in more detail? Please indicate the time period on the X-axis
In line no 160: Why is the plate being left to dry for two hours after applying the crystal violet?
Vibrio parahaemolyticus primarily relies on its virulence factors for its pathogenicity. If possible, kindly assess the indole production using your hit compounds.
In the conclusion section or end of the manuscript: it is suggested to add a summary of the research including significant findings and also some qualitative results that could enhance the readability of this research
